# SMART: Sentences as Basic Units for Text Evaluation

**Reinald Kim Amplayo, Peter J. Liu, Yao Zhao, Shashi Narayan**
Google Research
{reinald, peterjliu, yaozhaoyz, shashinarayan}@google.com

## Abstract

Widely used evaluation metrics for text generation either do not work well with longer texts or fail to evaluate all aspects of text quality. In this paper, we introduce a new metric called SMART to mitigate such limitations. Specifically, we treat sentences as basic units of matching instead of tokens, and use a sentence matching function to *soft*-match candidate and reference sentences. Candidate sentences are also compared to sentences in the source documents to allow grounding (e.g., factuality) evaluation. Our results show that system-level correlations of our proposed metric with a model-based matching function outperforms all competing metrics on the SummEval summarization meta-evaluation dataset, while the same metric with a string-based matching function is competitive with current model-based metrics. The latter does not use any neural model, which is useful during model development phases where resources can be limited and fast evaluation is required. SMART also outperforms all factuality evaluation metrics on the TRUE benchmark. Finally, we also conducted extensive analyses showing that our proposed metrics work well with longer summaries and are less biased towards specific models.

## 1 Introduction

One major obstacle in the progress of text generation tasks (e.g., document summarization, long-form question answering, data-to-text generation, etc.) is automatic evaluation. Traditionally, automatic metrics that rely on discrete token-level matching such as ROUGE (Lin, 2004) and BLEU (Papineni et al., 2002) have been utilized to check whether system outputs are of high quality across four dimensions (Kryscinski et al., 2019; Fabbri et al., 2021): coherence, factuality, fluency, and informativeness. These metrics do not correlate well with human judgments on all four dimensions of text quality (Fabbri et al., 2021). Because of this, the evaluation is usually coupled with human elicitation studies that ask humans to rate texts. These studies can be expensive and nearly impossible to reproduce.

More recently, pretrained language models are leveraged for automatically evaluating system-generated texts (Zhang* et al., 2020; Sellam et al., 2020; Yuan et al., 2021), which have shown improvements on correlation with human judgments. Nevertheless, both ROUGE and LM-

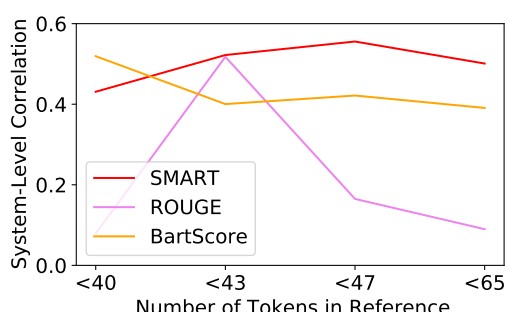

Figure 1: Kendall tau system-level correlations of ROUGE and SMART averaged over four dimensions of summary quality as the number of tokens increases. Summaries are from CNN/DM (Hermann et al., 2015) and human annotations are from SummEval (Fabbri et al., 2021). Each bucket in the x-axis contains equal number of data points. More details in Section A.2.

based metrics have three major drawbacks. Firstly, these metrics are not good at evaluating long and multi-sentence texts. Figure 1 illustrates system-level rank correlations of ROUGE in different text lengths, which shows that after a certain length, ROUGE drastically decreases its evaluative power. By design, ROUGE is also not robust to evaluating possibly shuffled information in long

outputs, hurting its performance on evaluating coherence. On the other hand, LM-based metrics such as the state-of-the-art BARTScore (Yuan et al., 2021), are constrained to the length limitation of the pretrained LM used, thus they are not able to evaluate outputs longer than this limit. And secondly, most of these metrics only use reference texts during evaluation. This restricts the capability of the metrics from evaluating dimensions of text quality that requires grounding to the source. Yuan et al. (2021) suggested to use the source document during evaluation, however their evaluation is still limited to short documents because of length limitations in LMs.

In this paper, we propose an automatic metric called SMART (**S**entence **MA**tching for **R**ating **T**ext)[1]. SMART is motivated by the pyramid method of human evaluation for summarization (Nenkova et al., 2007), where they transform text into semantic content units (SCUs), or sentences that contain a single fact. Since this kind of transformation cannot be done automatically, we use sentences as a proxy to SCUs, and treat them as basic units of matching instead of tokens. This additionally allows the metric to effectively support long and multi-sentence texts. Since sentences most likely do not have exact matches, we use a soft-matching function that returns a matching score between 0 and 1, given a pair of sentences. Moreover, to allow grounded evaluation, we also include the source in the calculation of the metric. Similar to ROUGE, we introduce multiple SMART versions using sentence n-gram overlap and longest common subsequence.

Our experiments show that SMART with BLEURT (Sellam et al., 2020) as a soft-matching function outperforms all the competing approaches on all four dimensions of quality in the SummEval dataset (Fabbri et al., 2019). We also show that SMART with T5-ANLI (Honovich et al., 2022) outperforms all competing factuality-based evaluation metrics on the TRUE benchmark (Honovich et al., 2022). Moreover, a faster variant of SMART, which does not use any neural model for text matching, shows competitive correlations with human judgments. Finally, our extensive analyses show that SMART works better with longer summaries and is less biased towards specific models.

## 2 RELATED WORK

Evaluation in conditional generation tasks such as machine translation and document summarization is a long-standing problem. Traditionally, evaluation involves human elicitation studies that score texts based on different metrics of quality, such as adequacy, fidelity, and fluency in machine translation (Hovy, 1999), and coherence, conciseness, fluency, readability, and content relevance in summarization (Mani, 2001; Nenkova et al., 2007). Automatic metrics based on token n-gram matching have been developed to replace these expensive and time-consuming studies, in which ROUGE (Lin, 2004) and BLEU (Papineni et al., 2002) are most widely used in summarization and translation, respectively. Several extensions to token n-gram matching have been proposed, such as using paraphrases, synonyms (Lavie & Agarwal, 2007), and word embeddings (Ng & Abrecht, 2015) to handle cases that are semantically equivalent, and downweighting common n-grams to focus more on salient ones (Vedantam et al., 2015). Popović (2015) instead use character-level n-gram matching to also match words that are conjugated differently and support morphologically rich languages.

With the introduction and success of pretrained language models such as BERT (Devlin et al., 2019) and BART (Lewis et al., 2020), evaluation metrics that leverage them have been proposed. BERTScore (Zhang* et al., 2020) leverages contextualized token embeddings from BERT and obtains pairwise matching of tokens from reference and system summaries. MoverScore (Zhao et al., 2019) extends BERTScore by instead having many-to-one soft alignments using Word Mover's Distance (WMD; Kusner et al., 2015). BLEURT (Sellam et al., 2020) fine-tunes BERT to predict human scores with large-scale synthetic training data. BARTScore (Yuan et al., 2021) uses BART and treats evaluation as a text generation problem, using likelihood of predicting the system summary given the source document or the reference summary. Clark et al. (2019) and Zhao et al. (2019) also explored sentence-level matching with WMD using (contextualized) sentence embeddings, however they show no concrete improvements over other model-based metrics (Fabbri et al., 2021). In contrast, we show that our metric correlates better with human judgments than all competing models.

Factuality in summarization (Falke et al., 2019; Maynez et al., 2020) is usually evaluated separately since most automatic metrics are focused on informativeness and do not include the source document in the metric calculation. Factuality-specific metrics can be divided into three approaches: natural

---

[1]`github.com/google-research/google-research/tree/master/smart_eval`

language inference (NLI) based approaches (Falke et al., 2019; Maynez et al., 2020; Laban et al., 2022), where a summary is considered factual if all its facts are entailed by the source document, model-based approaches (Kryscinski et al., 2020; Deng et al., 2021), where a model is trained to detect factual errors in the summary, and question answering (QA) based approaches (Durmus et al., 2020; Wang et al., 2020; Honovich et al., 2021), where questions generated in a factual summary should be answerable using the source. While we also compare correlations of automatic metrics with human judgments on factuality, the goal of our work is to find holistic metrics for evaluation that can also effectively evaluate other dimensions of text quality. Our results show that other dimensions such as coherence, fluency, and informativeness also benefit in the use of the source documents in the metric. Finally, we also show that our metric improves the state-of-the-art of factual consistency evaluation based on the TRUE benchmark Honovich et al. (2022).

## 3 PROBLEM DEFINITION

We primarily use document summarization – the task of generating concise and accurate summaries of input documents (Mani, 2001) – to explain our metric, but the proposed metric can be easily adapted to other tasks, as shown in Section 5.2.

Let $\mathcal{S}$ be a list of source documents, $\mathcal{C}$ be a list of summaries generated for $\mathcal{S}$ by a candidate system, and $\mathcal{R}$ be a list of reference summaries produced by human annotators for $\mathcal{S}$. Note that $\mathcal{R}$ can be a nested list, i.e., for each example, there can be multiple references. Moreover, let $\mathcal{Q}$ be a list of dimensions of summary quality, and let $\mathcal{H}_q$ be a list of human-annotated scores for $\mathcal{C}$ in terms of a certain summary quality $q$. For each summary quality $q \in \mathcal{Q}$, the problem is to devise an evaluation metric $f_q(\mathcal{S}, \mathcal{R}, \mathcal{C})$ that outputs a list of scores that correlates well with $\mathcal{H}_q$. Note that, unlike most of previous work on summarization evaluation (Lin, 2004; Clark et al., 2019; Bhandari et al., 2020; Fabbri et al., 2021), we also take into account source documents $\mathcal{S}$ when calculating metric $f_q(\cdot)$. This ensures that the metric can evaluate quality dimensions that require looking at the source.

We define the list of summary quality $\mathcal{Q}$ as the following four dimensions of summary quality, defined as follows (based on definitions in Fabbri et al., 2021 and Yuan et al., 2021):

- **Coherence**: The summary should be well-structured and well-organized. The summary should not just be a heap of related information, but should build from sentence to sentence to a coherent body of information about a topic.
- **Factuality**: The summary should only contain statements that are entailed by the source document. The summary should not contain hallucinated facts that either do not exist in the source or contradict facts from the source.
- **Fluency**: The summary should have no formatting problems, capitalization errors or obviously ungrammatical sentences (e.g., fragments, missing components) that make the text difficult to read.
- **Informativeness**: The summary should include only important information from the source document. The summary should not include redundant information and information that are considered excessive and non-salient.

## 4 SENTENCE MATCHING FOR RATING TEXT

We now describe our proposed metric, SMART (**S**entence **MA**tching for **R**ating **T**ext), which has two key ideas. Firstly, we treat sentences as basic units of matching between system and reference summaries, instead of tokens. At sentence-level, exactly matching sentences are most likely non-existent (in datasets with abstractive reference summaries), thus we instead utilize soft-matching functions to compare sentences. Similar to ROUGE, we present two types of SMART: n-gram overlap (SMART-N) and longest common subsequence (SMART-L). Secondly, SMART allows to compare the candidate system summary with *both* the reference summary and the source document. This is particularly important when evaluating dimensions of summary quality that rely on the source document such as factuality.

**SMART-N** In order for SMART to work, summaries should be split into sentences. Let $R = [r_i]$ and $C = [c_j]$ be the sequence of sentences of the reference and the candidate system summary.

```python
def soft_lcs(X, Y):
  lcs = [[0] * (len(Y)+1)] * (len(X)+1)
  for i in range(len(X)+1):
    for j in range(len(Y)+1):
      if i != 0 and j != 0:
        m = match(X[i], Y[j])
        lcs[i][j] = max(lcs[i-1][j-1]+m, lcs[i-1][j]+m, lcs[i][j-1])
  return lcs[-1][-1]
```

Figure 2: Python pseudocode of the soft version of Longest Common Subsequence (Soft-LCS) given two sets of summary sentences X and Y.

SMART-N finds pairs of sentence n-grams in $R$ and $C$ that have the highest matching scores given by a sentence matching function that returns a score between 0 and 1 (detailed description in Section A.1).

Formally, given $N$ as the number of sentences in the sentence n-gram, SMART-N can be calculated as follows:

$$prec' = \sum_{j=1}^{|C|-N+1} \max_{r_i \in R; i \leq |R|-N+1} \left[ \sum_{n=0}^{N-1} \texttt{match}(c_{j+n}, r_{i+n})/N \right] \quad (1)$$

$$prec = prec'/(|C| - N + 1) \quad (2)$$

$$rec' = \sum_{i=1}^{|\mathcal{R}|-N+1} \max_{c_j \in C; j \leq |C|-N+1} \left[ \sum_{n=0}^{N-1} \texttt{match}(r_{i+n}, c_{j+n})/N \right] \quad (3)$$

$$rec = rec'/(|R| - N + 1) \quad (4)$$

$$f = 2 * prec * rec/(prec + rec) \quad (5)$$

where $\texttt{match}(\cdot, \cdot)$ is the sentence matching function, $prec$, $rec$, and $f$ are precision, recall, and f-measure, respectively. Note that unlike ROUGE, the numerators of precision and recall are different due to the use of a soft-matching function, thus they are calculated separately. In our experiments, we used SMART-1 and SMART-2[2], but SMART-N can be easily extended to work with larger $N$s.

**SMART-L**  SMART-L is essentially the Longest Common Subsequence (LCS) of sentences in the reference and the candidate system summary. However, the original LCS algorithm requires an exact match to work. We instead use a *soft* version of LCS, where the task is defined as: Given two sequences $X = [x_i]$ and $Y = [y_j]$ and a matching function $\texttt{match}(x_i, y_j)$, find two *soft*-subsequences $x_{i_1}, ..., x_{i_l}$ and $y_{j_1}, ..., y_{j_l}$ of length $l$ with $i_{k-1} \leq i_k \leq i_{k+1}$ and $j_{k-1} \leq j_k \leq j_{k+1}$, maximizing the sum $\sum_{k=1}^{l} \texttt{match}(x_{i_k}, y_{j_k})$.

Unlike normal subsequences, soft-subsequences allow repetition of sentences as long as they do not go back to previous sentences (hence the use of $\leq$ operator). This relaxation helps in cases where the meaning of a sentence on one side spans over multiple sentences on the other side. Furthermore, Soft-LCS is similar but different from a simple sequence alignment problem since the weight of the match depends on *both* the positions of the items in the sequence and the items themselves. It is a less-restricted version of the Heaviest Common Subsequence (HCS; Jacobson & Vo, 1992) since the matching function is relaxed to allow the use of a soft match (which is essentially an exact mismatch) in the subsequence.

It turns out that Soft-LCS can be solved using a dynamic programming algorithm similar to that of LCS, which is illustrated as a pseudocode in Figure 2. The main difference is that since we do not require an exact match, we always take the maximum among three cases: (1) choosing to soft-match $x_i$ and $y_j$, (2) choosing to soft-match $x_i$ and $y_{j-1}$, and (3) choosing to skip $x_i$.

---

[2]In special cases where either the candidate or the reference is a single-sentence summary, a normal implementation of SMART-2 would return zero since one of the summaries would have zero sentence bigrams. To mitigate this issue, we pad summaries with a blank sentence on both sides when calculating SMART-2. This ensures that we get a non-zero score for single-sentence summaries. In fact, SMART-2 reduces to SMART-1 in this case.

Given the Soft-LCS function, we can then calculate SMART-L as follows:

$$prec' = \texttt{soft-lcs}(C, R) \tag{6}$$

$$prec = prec'/|C| \tag{7}$$

$$rec' = \texttt{soft-lcs}(R, C) \tag{8}$$

$$rec = rec'/|R| \tag{9}$$

$$f = 2 * prec * rec/(prec + rec) \tag{10}$$

**Comparing with Source** Some dimensions of summary quality require access to source to be effectively evaluated. To cover those dimensions, SMART also compares the candidate system summary with the source, in addition to comparison with the reference summary. Let $S = [s_k]$ be the sequence of sentences of the source document. SMART that uses both source and reference is calculated as follows. We first calculate two SMART scores that (1) compares candidate system summary $C$ with reference summary $R$, and (2) compares $C$ with source document $S$. Then, we aggregate the scores by taking their maximum. For example, $\texttt{SMART-N}(S, R, C)$ is calculated as:

$$\texttt{SMART-N}(S, R, C) = \max(\phantom{}$$
$$\texttt{SMART-N}(S, C), \texttt{SMART-N}(R, C)) \tag{11}$$

**Multiple References** Finally, when there are multiple reference summaries, we calculate SMART for each reference, and aggregate them by taking their maximum, as also commonly done in previous work (Fabbri et al., 2021). This is intuitive since the candidate system summary only needs to match with at least one of the reference.

**Shorter Acronym** We use the following template to describe SMART variants in a space-efficient manner: `S[1|2|L]-m`, where `m` is the sentence matching function of choice. For example, SMART-1 with a BLEURT (Sellam et al., 2020) matching function can be shortened into S1-BLEURT.

## 5 EXPERIMENTS AND RESULTS

**Sentence Matching Functions** One advantage of SMART is that it is easily extensible by changing the matching functions depending on the task. We investigated and compared six different matching functions, three model-based and three string-based, in Sections A.1 and A.5 of the Appendix and use the ones that perform the best for each task. Specifically, for document summarization, we use BLEURT (Sellam et al., 2020) as the best model-based function and CHRF (Popović, 2015) as the best string-based function. For factual consistency, we use ANLI (Honovich et al., 2022) as the matching function. Using these matching functions for the corresponding tasks is intuitive; BLEURT has been specifically trained to rank texts using objectives that correspond to multiple dimensions of text quality, while ANLI is an NLI classifier that outputs a binary prediction (entailed or not).

**Implementation Details** The sentence matching functions are implemented as follows. For BLEURT (Sellam et al., 2020), we used the `BLEURT-20` checkpoint[3] suggested by the authors which also supports non-English languages. For ANLI, we used the same implementation as in Honovich et al. (2022), where T5-11B is fine-tuned with 25K training steps on ANLI (Nie et al., 2020), treating both contradiction and neutral pairs as not entailed. Sentences are split using `nltk`[4]. For the document summarization experiments, we report f-measure scores whenever available, such as in ROUGE, BERTScore, and SMART. For the factual consistency experiments, we report precision scores. We also report the version of SMART that considers both source and reference as in Eq 11.

We tested our metric in two different tasks: document summarization and factual consistency. We discuss them in detail in the next sections.

### 5.1 DOCUMENT SUMMARIZATION

**Dataset and Evaluation** We conducted experiments on the SummEval dataset (Fabbri et al., 2021), a document summarization meta-evaluation suite consisting of summaries from the CNN/DM dataset

---

[3] https://github.com/google-research/bleurt
[4] https://pypi.org/project/nltk/

(Hermann et al., 2015). Annotation is done in two stages and using experts to ensure high quality and high agreement across annotators. There are 1600 data points (16 systems × 100 examples) in total, each of which includes a score between 1 to 5 for each of the four dimensions of summary quality, which represents the average score given by three experts. Each data point also includes 11 reference summaries: the original summary from the CNN/DM dataset and 10 human-written abstractive summaries from Kryscinski et al. (2020). For evaluation, we use system-level correlation using Kendall tau, where we first take the average score for each system and take the correlation.[5]

**Comparison Metrics**    We compared SMART with four types of metrics: source-free and source-dependent string-based metrics, and source-free and source-dependent model-based metrics.

*Source-free string-based metrics* include: (1-3) ROUGE-1/2/L (Lin, 2004) measures token-level overlap between reference and output summaries; (4) BLEU (Papineni et al., 2002) measures token-level overlap between reference and output summaries with a focus on precision; and(5) CHRF (Popović, 2015) measures character-based n-gram overlap between reference and output summaries.

*Source-dependent string-based metrics* include: (6-9) src-ROUGE-1/2/L and src-CHRF, versions of ROUGE and CHRF that compare the output summary with the source instead of with the reference; and (10-12) S1/2/L-CHRF, our best SMART metric using a string-based matching function, as shown in §A.1 of the Appendix.

*Source-free model-based metrics* include: (13) BERTScore (Zhang* et al., 2020): A metric that relies on BERT(-like) models (Devlin et al., 2019) and computes an aggregation of the token-level similarity scores; (14) MoverScore (Zhao et al., 2019): measures the semantic distance between BERT n-gram embeddings of reference and candidate summaries using Word Mover's Distance (WMD; Kusner et al., 2015); (15) BLEURT (Sellam et al., 2020): finetunes BERT using a combination of real and synthetic training data with gold-/silver-standard human judgment scores; and (16) Sentence Mover's Similarity (SMS; Clark et al., 2019): uses an extension of WMD that works with sentences instead of tokens.

*Source-dependent model-based metrics* include: (17) src-BLEURT, a version of BLEURT that compare the output summary with the source instead of with the reference; (18) PRISM (Thompson & Post, 2020): leverages a zero-shot paraphrasing model and uses probabilities from force-decoding the candidate summary given the source as input; (19) $Q^2$ (Honovich et al., 2021): employs question generation and question answering models and checks whether answers from the summary are entailed by answers from source; (20) ANLI (Honovich et al., 2022): fine-tunes T5 (Raffel et al., 2020) using the ANLI dataset (Nie et al., 2020) to produce an entailment score given the source as premise and the summary as hypothesis; (21-22) BARTScore(+CNN) (Yuan et al., 2021): evaluates text using probabilities from force-decoding the candidate summary given the source as input using BART (Lewis et al., 2020) without (with) finetuning with CNN/DM summarization dataset (Hermann et al., 2015); and (23-25) S1/2/L-BLEURT: Our best SMART metric using a model-based matching function, as shown in §A.1 of the Appendix.

**Results**    Table 1 reports the system-level correlations of different metrics for each quality dimension. For all quality dimensions, SMART with BLEURT matching function has the highest correlation, where SL-BLEURT evaluates coherence and informativeness better, and S1-BLEURT and S2-BLEURT evaluate factuality and fluency better, respectively. On average, SL-BLEURT performs best, followed by S1-BLEURT and S2-BLEURT, all three of which outperforming BARTScore+CNN, which is finetuned on the same summarization dataset as SummEval. S2-BLEURT also performs comparably with $Q^2$ in factuality evaluation. Given that each of the SMART metrics are better at evaluating different quality dimensions, it is therefore recommended to use them as a set, similar to how ROUGE metrics are used. We can also see in the table that source-dependent metrics are better than source-free ones, signifying the importance of the use of source documents during summary evaluation. However, just simply using the source does not improve the metric; src-ROUGE-1/2/L, src-CHRF, and src-BLEURT obtain worse correlation scores compared to their reference-only counterparts. Among source-free metrics, SMS performs the best, showing the superiority of sentence-

---

[5]While they claimed to report system-level correlation, the BARTScore paper (Yuan et al., 2021) actually calculated *summary*-level correlation (Louis & Nenkova, 2013), where they first get correlation for each system and then take the average. Since we use evaluation metrics to rank *systems*, we report system-level correlation following Fabbri et al. (2021).

| Metric | Coherence | Factuality | Fluency | Informativeness | $\mu$ |
|--------|-----------|------------|---------|-----------------|-------|
| *Source-free String-based Metrics* | | | | | |
| ROUGE-1 | .350 | .550 | .527 | .583 | .503 |
| ROUGE-2 | .233 | .600 | .494 | .433 | .440 |
| ROUGE-L | .117 | .117 | .259 | .350 | .211 |
| BLEU | .217 | .050 | .326 | .383 | .244 |
| CHRF | .350 | .617 | .561 | .550 | .519 |
| *Source-dependent String-based Metrics* | | | | | |
| src-ROUGE-1 | .000 | .467 | .226 | .200 | .223 |
| src-ROUGE-2 | .067 | .500 | .293 | .267 | .282 |
| src-ROUGE-L | .117 | .550 | .310 | .317 | .323 |
| src-CHRF | .050 | .517 | .276 | .250 | .273 |
| S1-CHRF | .300 | **.733** | .494 | .500 | .507 |
| S2-CHRF | .300 | .700 | .460 | .433 | .473 |
| SL-CHRF | .367 | **.733** | .494 | .500 | .523 |
| *Source-free Model-based Metrics* | | | | | |
| BERTScore | .333 | -.030 | .142 | .200 | .161 |
| MoverScore | .217 | -.050 | .259 | .350 | .194 |
| BLEURT | **.533** | .200 | .410 | .467 | .403 |
| SMS | .267 | .600 | .360 | .400 | .407 |
| *Source-dependent Model-based Metrics* | | | | | |
| PRISM | .233 | .600 | .360 | .367 | .390 |
| ANLI | .250 | .583 | .544 | .517 | .473 |
| $Q^2$ | .250 | **.750** | .577 | .450 | .507 |
| src-BLEURT | .317 | .717 | .510 | .483 | .507 |
| BARTScore | .350 | .617 | .494 | .450 | .478 |
| BARTScore+CNN[†] | **.550** | .317 | .594 | .583 | .511 |
| S1-BLEURT | .433 | .667 | **.644** | .667 | .603 |
| S2-BLEURT | .417 | **.750** | **.628** | .583 | .594 |
| SL-BLEURT | **.567** | .567 | .611 | **.733** | .619 |

Table 1: Kendall tau system-level correlation of different metrics on the SummEval dataset. Correlations of metrics not significantly outperformed by any other metric (using William's Test; Graham & Baldwin, 2014) for that specific dimension are **boldfaced**. We also show the average scores in the $\mu$ column, where the best values for each block are underlined. Note that BARTScore+CNN uses BART that is fine-tuned on CNN/DM, the same data source as SummEval, thus direct comparison with the other metrics is not possible.

level matching metrics against the token-level ones. Among string-based metrics, SL-CHRF performs the best, with a competitive system-level correlation when compared with BARTScore+CNN on average. This shows that SMART achieves comparable evaluation power with previous LM-based metrics even without using any pretrained language models.

**Ablation Studies** We present various ablation studies on the different components of SMART in Table 2. For simplicity, we report on SMART-X, an average of SMART-1, SMART-2 and SMART-L.[6] The first block contains SMART-X that uses only either precision or recall, both of which have substantially lower system-level correlation than f-measure. The recall variant performs better, which follows results from traditionally recall-oriented evaluation metrics in summarization (Lin, 2004).

The second block contains different ways to aggregate SMART scores that compare candidate summaries (1) to the source documents and (2) to the reference summaries. When using only one of the two scores (i.e., reference- or source-only), SMART scores significantly perform worse, which implies that using both source and reference is necessary for SMART to work. We also tried aggregation through taking the average or the minimum, both of which perform worse than taking the maximum as in Eq 11. Taking the minimum notably does not increase the correlation, and this

---

[6]From here on, we use `X` to correspond to the average of the three `[1|2|L]` variants of ROUGE or SMART.

| Component | Coherence | Factuality | Fluency | Informativeness | $\mu$ |
|---|---|---|---|---|---|
| Sx-BLEURT | .450 | .750 | .661 | .650 | .628 |
| *using precision or recall* | | | | | |
| precision | .083 | .017 | .259 | .350 | .177 |
| recall | .283 | .683 | .410 | .383 | .440 |
| *using a different source/reference aggregation* | | | | | |
| ref-only | -.100 | -.300 | -.092 | -.033 | -.131 |
| src-only | .267 | .700 | .460 | .433 | .465 |
| average | .400 | .750 | .577 | .600 | .582 |
| minimum | .267 | .700 | .427 | .467 | .465 |
| maximum w/o sentence-level | .550 | .550 | .494 | .517 | .528 |
| *using a different reference summary* | | | | | |
| ref-only (CNN/DM) | -.100 | -.300 | -.059 | -.033 | -.123 |
| ref-only (best system) | .367 | .100 | .410 | .500 | .344 |
| max (CNN/DM) | .500 | .767 | .711 | .667 | .661 |
| max (best system) | .667 | .467 | .745 | .867 | .686 |

Table 2: Ablation study. Kendall tau system-level correlation of Sx-BLEURT when one component is set to a different configuration.

is because it restricts giving high scores to summaries that match *both* the source and the reference, which is not necessary. Finally, removing the sentence-level mechanism of SMART also reduces the overall correlation score.

One interesting finding is that using only the reference summaries to calculate SMART gives negative correlation scores on all dimensions. This can be explained in two-folds. Firstly, human annotations in SummEval is done by only looking at the source document, which means they are not biased towards summaries that look closer to the given reference summaries. And secondly, Fabbri et al. (2021) showed that CNN/DM summaries are worse than system summaries in terms of human ratings. Given these, we further investigated using different reference summaries for SMART. Specifically, we tried (1) replacing the set of reference summaries from SummEval into the original summary from CNN/DM, and (2) treating the best system summary according to the average human score as the reference. As can be seen in the third block of Table 2, CNN/DM reference summaries correlate negatively on all dimensions. When using the best system summary as the reference, SMART now obtains positive correlations across all dimensions. The correlation further improves when SMART uses both the source and the new reference.

**Further Analyses**  We provide several analyses in the Appendix to further understand our metrics. In §A.2, we show that SMART works better with longer summaries, and in §A.3, we also show that it is less biased towards specific models compared to other competing metrics. Finally, in §A.4, we show that source-only SMART works well as a reference-free metric and correctly ranks systems that popular metrics such as ROUGE (Lin, 2004), BERTScore (Zhang* et al., 2020), and QAEval (Deutsch et al., 2021) cannot, according to Goyal et al. (2022).

## 5.2 Factual Consistency

**Dataset and Evaluation**  We conducted experiments on the TRUE benchmark (Honovich et al., 2022), a meta-evaluation suite for factuality, consisting of source and candidate texts from different kinds of text generation tasks: abstractive summarization, dialogue generation, and paraphrase detection. We note that there is no reference text given and the candidate text is compared only to the source. In this benchmark, factuality scores are binary (positive/negative, in contrast to the five-scale scoring in SummEval), and the generated text needs to be fully factual to get a positive score. Given this, the metrics are evaluated using the Received Operating Characteristic Area Under the Curve (ROC AUC) to remove the necessity of setting a specific decision threshold.

| Task | 3-Ensemble | 2-Ensemble | | Single Metrics | | | |
| --- | --- | --- | --- | --- | --- | --- | --- |
| | ANLI + Q$^2$ + SC$_{ZS}$ | ANLI + SX-ANLI | ANLI + Q$^2$ | SX-ANLI | ANLI | SC$_{ZS}$ | Q$^2$ |
| FRANK | 91.2 | 91.1 | 89.6 | 89.4 | 89.4 | 89.1 | 87.8 |
| SummEval | 82.9 | 82.2 | 80.7 | 80.6 | 80.5 | 81.7 | 78.8 |
| MNBM | 76.6 | 77.0 | 75.6 | 73.0 | 77.9 | 71.3 | 68.7 |
| QAGS-C | 87.7 | 85.3 | 86.0 | 81.7 | 82.1 | 80.9 | 83.5 |
| QAGS-X | 84.8 | 84.0 | 81.8 | 77.2 | 83.8 | 78.1 | 70.9 |
| BEGIN | 86.2 | 84.9 | 85.7 | 85.0 | 82.6 | 82.0 | 79.7 |
| Q$^2_{dataset}$ | 82.8 | 81.5 | 83.0 | 82.4 | 72.7 | 77.4 | 80.9 |
| DialFact | 90.4 | 90.3 | 89.4 | 90.4 | 77.7 | 84.1 | 86.1 |
| PAWS | 91.2 | 95.7 | 90.5 | 98.3 | 86.4 | 88.2 | 89.7 |
| Average | **86.0** | **85.8** | 84.7 | **84.2** | 81.5 | 81.4 | 80.7 |

Table 3: ROC AUC results for the different metrics on the development set of the TRUE benchmark. Results can be from ensembles of multiple metrics, and are grouped by the number of metrics used in the ensemble. The highest average score for each group is **boldfaced**.

**Comparison Metrics**   The best matching function for factual consistency is ANLI, as shown in §A.5 of the Appendix. Since there is no reference text, we used the version of SMART that compares only with the source. Moreover, we aggregated S1/2/L-ANLI into SX-ANLI by taking their mean for brevity (results of individual metrics are reported in §A.5). We compared (1) SX-ANLI with three factuality-specific evaluation metrics: (2) Q$^2$ (Honovich et al., 2021); (3) ANLI (Honovich et al., 2022); and (4) SC$_{ZS}$ (Summary Consistency; Laban et al., 2022) that splits the source and candidate texts into sentences, computes entailment probabilities on all sentence pairs, and aggregates them into a single score. The last metric is similar to S1-ANLI, but with two key differences: (a) The aggregation is different; and (b) Training is done using both NLI and fact verification datasets.

Finally, following Honovich et al. (2022) we also compared three ensembles of multiple metrics by averaging their scores: (5) ANLI + Q$^2$, (6) ANLI + SX-ANLI, and (7) ANLI + Q$^2$ + SZ$_{ZS}$.

**Results**   Table 3 shows the ROC AUC scores of different metrics. Among single metrics, SX-ANLI performs the best, substantially improving over the second best metric. Moreover, ensembling SX-ANLI with ANLI provides another boost, performing superiorly compared to the ANLI + Q$^2$ ensemble metric and comparably with the ANLI + Q$^2$ + SZ$_{ZS}$ ensemble metric. Our ensemble metric is also more cost-effective since it only requires a single model, while the other ensemble metric requires at least two more (e.g., question answering and generation models for Q$^2$).

## 6  CONCLUSIONS

In this paper, we proposed SMART, a new metric for evaluating generated text given a source and reference text. SMART makes use of a sentence-level soft-matching function to match sentences, which can either be string-based or model-based depending on the available resources. This function can easily be replaced with new and better ones, which allows us to create a better SMART metric. We provided two types of SMART based on n-gram overlap and longest common subsequence, and our extensive experiments showed that SMART evaluates document summaries better in terms of all four dimensions of summary quality: coherence, factuality, fluency, and informativeness. Our analyses also showed that SMART is better as summary length increases and is less biased than other competing metrics.

**Limitations and Future Work**   We acknowledge several limitations of our work. Firstly, SMART currently assumes single-source inputs, which makes it non-trivial to use for evaluating texts in tasks with multi-source inputs. Secondly, we plan to develop a system to automatically transform text into a list of semantic content units (SCUs), or sentences that contain a single fact. This would improve SMART by supporting texts with sentences that have more than one atomic fact.   And finally, we also plan to explore cross-lingual capabilities of SMART-BLEURT, and apply to tasks where the source and candidate summaries are of different languages (e.g., machine translation and cross-lingual summarization).

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

# A  APPENDIX

## A.1  SENTENCE MATCHING FUNCTIONS

The scores we get from SMART depend on the sentence matching function `match`. We investigate six different sentence matching functions widely used in both machine translation and document summarization. Specifically, we compare three string-based and three model-based matching functions. The former do not rely on accelerators (GPUs/TPUs) while the latter leverage pretrained neural models. Thus, we expect string-based matching functions to be inferior, but they are good alternatives for faster evaluation or when accelerators are not available.

**ROUGE (Lin, 2004)**    A popular document summarization evaluation metric, it measures the number of overlapping textual units. As with most summarization work, we explored three types of textual units: unigrams (ROUGE-1), bigrams (ROUGE-2), and longest common subsequence (ROUGE-L).

**BLEU (Papineni et al., 2002)**    A popular machine translation evaluation metric, it is a precision-focused metric that calculates n-gram overlap between two texts and also includes a brevity penalty.

**CHRF (Popović, 2015)**    Another machine translation evaluation metric, it calculates character-based n-gram overlap between system and reference sentences. Unlike ROUGE and BLEU which operate at the token level, CHRF is more effective especially in morphologically-rich languages as it does not require any tokenization, lemmatization, and stemming (Mathur et al., 2020; Kocmi et al., 2021).

**BERTScore (Zhang* et al., 2020)**    A model-based metric that leverages contextualized token embeddings from BERT(-like models). It computes similarity scores by aligning tokens from reference and candidate summaries, and token alignments are computed greedily to maximize cosine similarity.

**ANLI (Honovich et al., 2022)**    Another model-based metric mainly used to evaluate factuality, which uses T5 (Raffel et al., 2020) fine-tuned on the ANLI dataset (Nie et al., 2020) to produce a score between 0 (not entailed) and 1 (entailed) given a premise and a hypothesis. We use the source/reference as premise and the candidate summary as hypothesis.

**BLEURT (Sellam et al., 2020)**    A supervised model-based metric that uses BERT that is trained to predict human judgment scores using a small-scale dataset. To make it more robust, the model is first pretrained with a large-scale synthetic dataset. Moreover, it is optimized using several objectives including ROUGE, BLEU, BERTScore, and entailment. BLEURT has been shown to be effective in evaluating sentence match in machine translation, thus we expect it to be the better matching function among all matching functions.

One advantage of SMART is that it is easily extensible by changing the matching functions to better ones. This means that a more domain-specific matching function can be used for evaluation towards specific domains, or a better-performing sentence matching metric can be used to improve overall evaluation.

**SMART with Different Matching Functions**    We compare different variants of SMART using the six matching functions described in Section A.1. Table 4 shows their system-level correlations, where Coh, Fac, Flu, and Inf stand for coherence, factuality, fluency, and informativeness, respectively. Among string-based matching functions, CHRF performs the best in terms of average correlation, followed by BLEU. This shows that machine translation metrics are better sentence matchers. Among model-based matching functions, BLEURT performs the best by a large margin; SMART with BLEURT significantly outperforms all the other model-based variants on all dimensions of summary quality. We believe that this is because BLEURT is optimized to match sentences, as well as to predict ROUGE, BLEU, BERTScore, and entailment scores. Interestingly, ANLI as a matching function underperforms even in the factuality dimension. We posit that this is because sentences are passed to the matching function without their neighboring context.  Finally, when compared against the no-SMART variant (e.g., BLEURT vs. S1/2/L-BLEURT), SMART improves the correlation overall,

| Matching Function | SMART Type | Coherence | Factuality | Fluency | Informativeness | $\mu$ |
|---|---|---|---|---|---|---|
| *SMART with String-based Matching Functions* | | | | | | |
| ROUGE-1 | S1 | .233 | **.733**$^*$ | .494 | .500$^*$ | .490 |
| | S2 | .217 | .683 | .477 | .417 | .448 |
| | SL | .267 | .700 | .460 | .467 | .473 |
| | none | .350 | .550 | .527 | .583 | .503 |
| ROUGE-2 | S1 | .183 | .650 | .477 | .417 | .432 |
| | S2 | .183 | .650 | .477 | .417 | .432 |
| | SL | .217 | .650 | .444 | .383 | .423 |
| | none | .233 | .600 | .494 | .433 | .440 |
| ROUGE-L | S1 | .233 | **.733**$^*$ | .527$^*$ | .500$^*$ | .498 |
| | S2 | .217 | .683 | .477 | .417 | .448 |
| | SL | .267 | .700 | .460 | .467 | .473 |
| | none | .117 | .117 | .259 | .350 | .211 |
| BLEU | S1 | .300 | **.733**$^*$ | .527$^*$ | .467 | .507 |
| | S2 | .283 | .717 | .510 | .450 | .490 |
| | SL | .317 | .717 | .460 | .483 | .494 |
| | none | .217 | .050 | .326 | .383 | .244 |
| CHRF | S1 | .300 | **.733**$^*$ | .494 | .500$^*$ | .507 |
| | S2 | .300 | .700 | .460 | .433 | .473 |
| | SL | .367$^*$ | **.733**$^*$ | .494 | .500$^*$ | .523 |
| | none | .350 | .617 | .561 | .550 | .519 |
| *SMART with Model-based Matching Functions* | | | | | | |
| BERTScore | S1 | .317 | .683 | .561 | .517 | .519 |
| | S2 | -.017 | .383 | .276 | .183 | .207 |
| | SL | .383 | .683 | .527 | .583 | .544 |
| | none | .333 | -.030 | .142 | .200 | .161 |
| ANLI | S1 | .117 | .550 | .444 | .350 | .365 |
| | S2 | .133 | .533 | .360 | .333 | .340 |
| | SL | .117 | .483 | .343 | .350 | 323 |
| | none | .250 | .583 | .544 | .517 | .473 |
| BLEURT | S1 | .433 | .667 | **.644**$^*$ | .667 | .603 |
| | S2 | .417 | **.750**$^*$ | **.628**$^*$ | .528 | .594 |
| | SL | **.567**$^*$ | .567 | .611 | **.733**$^*$ | .619 |
| | none | **.533** | .200 | .410 | .467 | .403 |

Table 4: Kendall tau system-level correlation of variants of SMART using different matching functions. For each block, correlations of metrics not significantly outperformed by any other metric (using William's Test; Graham & Baldwin, 2014) for that specific dimension are marked with an asterisk (*). Those that are not significantly outperformed by all metrics are **boldfaced**. We also show correlation scores of the matching function without SMART, as well as average scores in the $\mu$ column, where the top three values for each block are underlined.

except for three matching functions: ROUGE-1/2 and ANLI. We believe this is because they are not effective as a sentence matching function. ROUGE-1/2 only matches unigrams/bigrams, in contrast to BLEU (where all n-grams up to $n = 4$ are considered collectively) and BertScore (where the n-grams are contextualized). ANLI is an NLI classifier that produces predictions that are close to 0 or 1, which is not ideal for ranking two systems with different degrees of errors.

## A.2  SMART WORKS WELL WITH LONGER SUMMARIES

We divided the datasets into four buckets based on the average number of tokens in the reference summary, where the first bucket contains the shortest reference summaries. For each bucket, we then calculated system-level correlation for all competing metrics. For each quality dimension, we report the relative increase in correlation with respect to ROUGE-X, which is illustrated in Figure 3. As can be seen in the figure, in general, all metrics perform better relative to ROUGE as the number of tokens

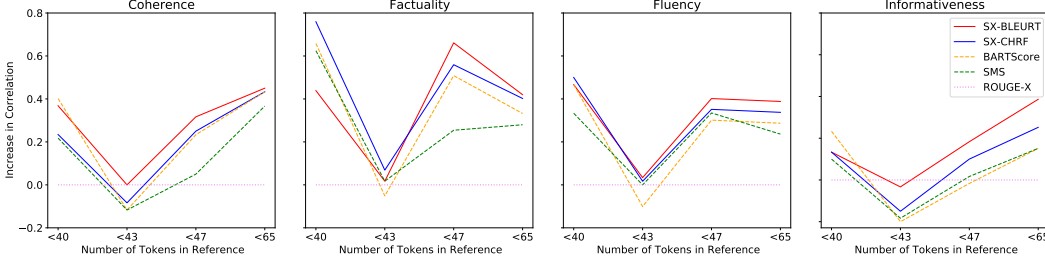

Figure 3: Length Analysis. Relative increase in Kendall tau system-level correlation with respect to ROUGE-X of four evaluation metrics for each length bucket (leftmost bucket has the shortest lengths).

| Metric | GovReport relevance | factuality | geomean | arXiv relevance | factuality | geomean |
|---|---|---|---|---|---|---|
| ROUGE-1 | 0.52 | -0.12 | 0.00 | 0.25 | -0.13 | 0.00 |
| ROUGE-2 | 0.44 | -0.11 | 0.00 | 0.16 | -0.13 | 0.00 |
| ROUGE-L | 0.39 | -0.11 | 0.00 | 0.17 | -0.15 | 0.00 |
| BERTScore | 0.38 | -0.04 | 0.00 | 0.18 | -0.10 | 0.00 |
| BLEURT | 0.06 | -0.08 | 0.00 | 0.37 | 0.00 | 0.01 |
| BARTScore | 0.25 | 0.06 | 0.12 | 0.12 | 0.24 | 0.17 |
| SX-ROUGE-1 | **0.72** | 0.06 | 0.21 | 0.27 | 0.52 | 0.37 |
| SX-CHRF | 0.24 | 0.39 | 0.27 | 0.30 | 0.48 | 0.38 |
| SX-BLEURT | 0.31 | 0.26 | **0.28** | 0.45 | 0.38 | **0.41** |
| SX-BLEURT-precision | -0.21 | **0.40** | 0.00 | 0.26 | **0.56** | 0.38 |
| SX-BLEURT-recall | 0.36 | -0.04 | 0.00 | **0.53** | 0.17 | 0.30 |

Table 5: Spearman rank correlation of different metrics on the GovReport and arXiv datasets with annotations provided in Koh et al. (2022). The values in the 'geomean' column are the geometric averages (we put zero if one value is non-positive) of the correlation scores. The best number for each column is **boldfaced**.

increases from 43 tokens, which shows that ROUGE is not suitable for long summary evaluation. Interestingly, ROUGE also underperforms in the first bucket, which means that it is also not good at evaluating short summaries. Among the competing metrics, SX-BLEURT (and SX-CHRF) correlate the best (and second best) when the there are more tokens in the source/reference.

We also compared SMART with different metrics on two long-form summarization datasets, arXiv (Cohan et al., 2018) and GovReport (Huang et al., 2021). We used the meta-evaluation suite curated in Koh et al. (2022), where they annotate scores in terms of relevance (measures whether a summary contains the salient information found in the reference) and factuality (assesses whether a summary is factually consistent with the source). Table 5 reports the Spearman rank correlation scores of competing metrics, including ROUGE-1/2/L, BERTScore, BLEURT, BARTScore, and SX-ROUGE-1/CHRF/BLEURT.[7] In terms of the geometric average of both relevance and factuality, SX-BLEURT performs the best among all metrics. All previous metrics except BARTScore only positively correlates with relevance. BARTScore is positively correlated with both relevance and factuality, however it is substantially worse than SMART. When dissecting SMART into its precision and recall variants, we see that the precision variant is correlated well with factuality while the recall variant is correlated well with relevance. This shows that SMART can be used as a single metric to evaluate both relevance and factuality, which can be useful for hill-climbing during model development.

---

[7] SX-ROUGE-1 has the best correlation among the SX-ROUGE-1/2/L variants.

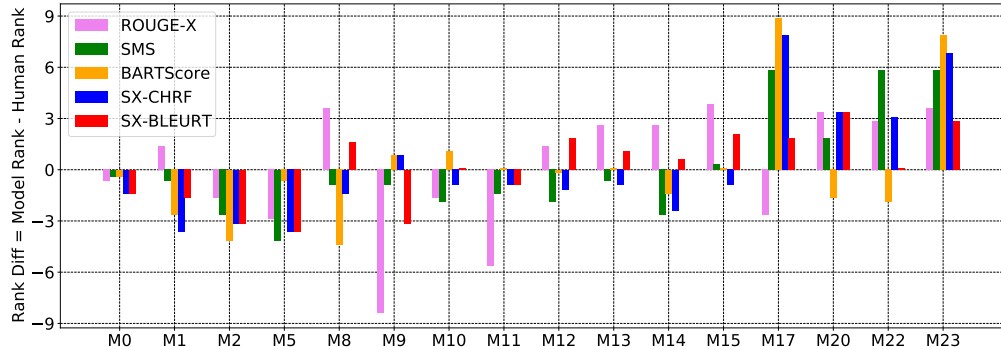

Figure 4: Bias Analysis. Difference in ranking between human scores and metric scores for each competing metric, averaged over all quality dimensions. A negative value means that the metric ranks the system higher. M0 to M23 in the x-axis correspond to the summarization models, where M0 to M5 are extractive and the rest are abstractive. See Fabbri et al. (2021) for detailed model descriptions.

|  | Metric | $\sigma$(Rank Diff) $\downarrow$ | Pairwise Acc. $\uparrow$ |
|---|---|---|---|
| Coh | ROUGE-X | 5.012 | 63.33 |
|  | SMS | 5.500 | 63.33 |
|  | BARTScore | 4.717 | 67.50 |
|  | SX-CHRF | 5.232 | 65.00 |
|  | SX-BLEURT | **4.228** | **72.50** |
| Fac | ROUGE-X | 4.228 | 70.00 |
|  | SMS | 2.915 | 80.83 |
|  | BARTScore | 2.915 | 80.83 |
|  | SX-CHRF | **2.062** | 86.67 |
|  | SX-BLEURT | 2.151 | **87.50** |
| Flu | ROUGE-X | 3.969 | 72.50 |
|  | SMS | 4.650 | 67.50 |
|  | BARTScore | 4.031 | 75.00 |
|  | SX-CHRF | 3.824 | 74.17 |
|  | SX-BLEURT | **2.646** | **83.33** |
| Inf | ROUGE-X | 3.841 | 75.00 |
|  | SMS | 4.783 | 70.00 |
|  | BARTScore | 4.500 | 72.50 |
|  | SX-CHRF | 4.198 | 75.00 |
|  | SX-BLEURT | **2.739** | **82.50** |

Table 6: The standard deviation of the difference in ranking between human and metric scores $\sigma$(Rank Diff) and the pairwise ranking accuracy of different metrics for different summary quality. Best scores are **boldfaced**.

## A.3 SMART IS LESS BIASED TOWARDS SPECIFIC MODELS

While we acknowledge that all automatic metrics are not perfect as shown in Table 1, their rankings should not be hugely different from human rankings. Moreover, they should not be biased towards a single summarization model. To this end, we get the difference in rankings given by humans and by the automatic metrics for each summarization model for each quality dimension. Figure 4 illustrates the resulting differences averaged over all dimensions of summary quality, in which we have two interesting observations. Firstly, all automatic metrics are in general biased towards ranking extractive systems higher. This suggests that a separate extractive and abstractive model evaluation is necessary using current automatic metrics. We leave exploration of metrics that are equally unbiased towards both extractive and abstractive for future work. Secondly, we found that BARTScore scores BART (M22 in Figure 4; Lewis et al., 2020) significantly higher than all the

| Model | Human | ROUGE-1 | CHRF | BLEURT | SX-ROUGE-1 | SX-CHRF | SX-BLEURT |
|---|---|---|---|---|---|---|---|
| | | | *BBC Dataset* | | | | |
| GPT3-D2 | 1st | 15.812 | 9.512 | 41.871 | 39.621 | 43.461 | 49.275 |
| T0 | 2nd | 10.617 | 5.641 | 39.952 | 29.440 | 30.210 | 42.747 |
| BRIO | 3rd | 10.964 | 5.700 | 39.200 | 31.070 | 31.030 | 41.852 |
| | | | *CNN Dataset* | | | | |
| GPT3-D2 | 1st | 24.357 | 16.020 | 42.946 | 40.115 | 40.382 | 48.911 |
| BRIO | 2nd | 22.128 | 13.564 | 42.950 | 35.582 | 31.800 | 43.436 |
| T0 | 3rd | 15.182 | 9.163 | 40.641 | 31.940 | 29.096 | 41.486 |

Table 7: Scores of three systems on two different datasets based on five different reference-free metrics. Human rankings are also shown for comparison. Values that are colored red are those that do not correspond to the human rankings.

other models (BARTScore of BART is $-1.398$ vs. $-2.303 \pm 0.313$ on average without BART). This problem is amplified when BARTScore is finetuned using the CNN/DM dataset ($-0.488$ vs. $-1.702 \pm 0.248$). This shows that using pretrained encoder-decoder models for summary evaluation induces bias towards summarization models finetuned on the same model.

In Figure 4, we can see that SX-BLEURT is the least biased since its rank differences with human scores are closer to zero. To quantitatively measure if the above statement is true, we use two measures. The first one is the standard deviation of the rank difference, where the score closest to zero can be considered the least biased. The second measure is the pairwise ranking accuracy, where for all pairs of system, we check whether human and metric rankings are equivalent. Table 6 shows these numbers, which show that SX-BLEURT has the lowest standard deviation of the rank difference and the highest pairwise rank accuracy across all quality dimensions. This entails that the metric is the least biased among the competing metrics.

## A.4 SMART IS EFFECTIVE EVEN WITHOUT REFERENCE SUMMARIES

In this section, we show that SMART does not require reference summaries to effectively evaluate system summaries. We make use of two sets of newswire data from BBC and CNN from Goyal et al. (2022). Each dataset consists of a source news article and three candidate summaries generated by (1) GPT3-D2: a GPT-3 model (Brown et al., 2020) which is trained to follow user instructions (Ouyang et al., 2022); (2) BRIO (Liu et al., 2022): a summarization model fine-tuned on both CNN/DM (Hermann et al., 2015) and XSum (Narayan et al., 2018) which currently achieves state-of-the-art results; and (3) T0 (Sanh et al., 2022): a prompt-based model fine-tuned on multiple tasks which include summarization datasets.

Goyal et al. (2022) showed that reference-dependent metrics such as ROUGE (Lin, 2004), BERTScore (Zhang* et al., 2020), and QAEval (Deutsch et al., 2021) do not correlate well with human judgments. We therefore compared *reference-free* versions of five metrics: ROUGE-1,[8] CHRF, BLEURT, and their SMART variants, where we calculate the scores by instead comparing the candidate summary with the source. Table 7 reports the scores of these metrics on the BBC and CNN datasets. As can be seen in the table, only SX-BLEURT is able to rank the systems correctly on both datasets, according to the given human rankings. All the other scores produce incorrect rankings on one dataset, however GPT3-D2 is ranked the highest in most cases for all metrics. This entails that using reference summaries may not be necessary (and can be detrimental) to evaluate system summaries, and our results demonstrate that SMART-BLEURT is better equipped than other metrics.

## A.5 SMART FOR ASSESSING FACTUAL CONSISTENCY

Table 8 reports the ROC AUC scores of different individual SMART metrics using three different kinds of matching functions: ANLI, BLEURT, and CHRF. Overall, SMART with ANLI performs significantly better than the other matching functions. This is expected since ANLI produces scores

---

[8]ROUGE-1, ROUGE-2, and ROUGE-L all have similar trends. We only report ROUGE-1 for brevity.

| Task | S1-ANLI | S2-ANLI | SL-ANLI | S1-BLEURT | S2-BLEURT | SL-BLEURT | S1-CHRF | S2-CHRF | SL-CHRF |
|---|---|---|---|---|---|---|---|---|---|
| FRANK | 90.5 | 89.9 | 90.4 | 83.8 | 84.8 | 82.9 | 83.3 | 84.0 | 83.2 |
| SummEval | 80.8 | 77.4 | 81.0 | 75.3 | 74.8 | 75.8 | 75.2 | 75.5 | 75.9 |
| MNBM | 73.0 | 73.0 | 73.0 | 67.9 | 68.0 | 67.9 | 68.0 | 68.0 | 68.0 |
| QAGS-C | 83.0 | 79.7 | 79.1 | 80.9 | 79.2 | 79.4 | 75.5 | 75.2 | 74.9 |
| QAGS-X | 77.2 | 77.2 | 77.2 | 61.6 | 61.6 | 61.6 | 59.0 | 59.0 | 59.0 |
| BEGIN | 85.0 | 84.8 | 85.0 | 86.7 | 86.8 | 86.7 | 83.6 | 83.5 | 83.6 |
| $Q^2_{dataset}$ | 82.3 | 82.5 | 82.2 | 69.6 | 70.5 | 69.6 | 60.7 | 62.0 | 60.7 |
| DialFact | 90.4 | 90.2 | 90.4 | 65.8 | 69.9 | 66.7 | 65.8 | 68.6 | 65.7 |
| PAWS | 98.3 | 98.3 | 98.3 | 66.4 | 66.3 | 66.4 | 66.3 | 66.2 | 66.3 |
| Average | 84.5 | 83.7 | 84.1 | 73.1 | 73.5 | 73.0 | 70.8 | 71.3 | 70.8 |

Table 8: ROC AUC results for the different metrics on the development set of the TRUE benchmark. The highest average score is **boldfaced**.

that are extremely polarized (i.e., close to 0 or 1) even for slightly nonfactual pairs, which is more appropriate for this task. BLEURT and CHRF, on the other hand, produce partial scores for such pairs, which make them more appropriate in ranking systems, such as in SummEval.

