# OpenReview forum: "SMART: Sentences as Basic Units for Text Evaluation"
_ICLR.cc/2023/Conference — ICLR 2023 poster_

### Official Review · Reviewer_7UHv · 2022-10-23

**Confidence:** 4
**Correctness:** 3
**Technical Novelty And Significance:** 3
**Empirical Novelty And Significance:** 3
**Recommendation:** 6

**Clarity, Quality, Novelty And Reproducibility:**

Clarity: the paper is overall clearly written.

Quality: there are some concerns about experiment design, as stated above.

Novelty: the idea of sentence-level soft-matching for summarization evaluation carries non-trivial novelty. For source inclusion, the idea is mostly imported from previous work.

Reproducibility: the method does not involve process with significant randomness (e.g. neural model training), and details of methods in the experiments (proposed and compared) are provided, therefore should be highly reproducible.

**Strength And Weaknesses:**

#### Strength

- Evaluation metrics for summarization is a highly important topic, especially because the currently most widely-used metrics like ROUGE has been found to correlate poorly with human evaluation.

- The proposed method has non-trivial novelty (mostly in the sentence-level soft-matching component), and has exhibited good performances in the experiments.


#### Weakness
- As stated above, the novelty mostly resides in the sentence-level soft-matching techniques. However, the ablation studies did not effectively verify the usefulness of it. The ablation studies showed the usefulness precision & recall, source & reference, and max aggregation, yet it is possible that the performance gain mostly comes from source inclusion but not sentence-level soft-matching. This is likely, given the poor performance of the "ref-only" setting and relatively better performance of "src-only", implying that source is more helpful than references. A valuable (and hopefully easy) study would be to use existing methods, like BLEURT, ANLI, etc., *with* source inclusion but *without* sentence-level soft-matching. If it consistently underperforms the full version of SMART, that verifies the usefulness of sentence-level soft-matching.

- The improvement of SMART is not always consistent; it does not seem to improve all methods. For example, based on Table 1 and Table 4, it seems to me that it only improves ROUGE-L, BLEU, BLEURT and BertScore, but hurts ROUGE-1/2 and ANLI. Although it's good to be able to improve on a current SOTA method and establish a new SOTA, it becomes questionable whether the method generalizes to future SOTA methods. (By the way, I think it would be better to put information in Table 1 also in Table 4 so that it's easier to compare.)


Below are my questions, not necessarily weaknesses:
- Some results in Table 2 are quite hard for me to understand. Mainly, SX-BLEURT ref-only gets extremely bad performance (negative correlation with human on all aspects), but BLEURT without SMART is not as bad. Also, even though ref-only has negative correlation, with source added, including the references can still largely improve the performance. How would you interpret these results?

- A more high-level question: how do you justify the idea of using sentences as units in summary evaluation? As I imagine, there can be cases where information is grouped differently among sentences in prediction and reference (e.g. predicted sentences = ["A", "BC"]; reference sentences = ["AB", "C"]. The prediction is perfect, but all sentence-pair matching scores are imperfect, so the given score will be imperfect). In such cases, using sentence as units will not be better than using tokens as units.


Below are some minor points:
- Many argued strengths of the method, such as good for longer summaries, less biases, good reference-free performances, etc., are only elaborated in the appendix.

- (typo) page 9, above "Results": "SZ_{ZS}" should be "SC_{ZS}"

**Summary Of The Paper:**

This paper proposes SMART, a new metric for conditional text generation (especially summarization) evaluation. The proposed method mainly consists of two components: sentence-level soft-matching and source inclusion. For sentence-level soft-matching, it splits the prediction and reference into separate sentences. The single sentence pair similarities are computed using existing metrics (e.g. BLEURT, CHRF, etc.) The sentence-level similarities are aggregated in ways inspired by ROUGE, leading to different method variations, SMART-N (based on sentence N-gram) and SMART-L (based on longest common sequence, LCS). For SMART-L, a dynamic programming algorithm is proposed to find the best soft-alignment between prediction sentences and reference sentences. For source inclusion, it essentially treats the source (input) text as another reference, and aggregate the source score with other reference scores by taking their maximum.

The authors conducted evaluations on two meta-evaluation datasets: SummEval, for general summary evaluation; and TRUE, for summary factual verification. Experiments show that using SMART on existing good-performing methods can further improve the performance and achieves overall the best performance compared to other methods. Ablation studies are also conducted to verify different components in the method, and provide more insights into the method behaviors.

**Summary Of The Review:**

This paper proposes SMART, a new metric for summarization evaluation based on a novel technique, sentence-level soft-matching, and source inclusion. Experiments show the good performance of the method. However, the study was not rigorously done to verify the usefulness of the novel part. Also, the improvement is not fully consistent among different based methods. The work is overall interesting, but more work should be done to make it more concrete and strong.

(Scores updated after rebuttal)

---

> ### Author Response · Authors · 2022-11-14
> **Response**
>
> We thank the reviewer for their valuable comments. We updated the paper, addressing the following concerns mentioned in the above review:
>
> 1. “It is possible that the performance gain mostly comes from source inclusion but not sentence-level soft matching.”
>
> We include versions of ROUGE-1/2/L, CHRF, and BLEURT that compare with the source in Table 1. In contrast to the reviewer’s argument, all of them perform worse than their SMART versions (see general response for a summary of the results).
>
> There are two possible reasons for this. Firstly, the source document is naturally longer than the reference summary, and as also argued in the paper, these metrics are not effective when the documents are longer. Secondly, the source document contains information that does not need to be in the summary. These metrics factor all this excess information, making the scores really low. The sentence-level computation in SMART provides a way to handle both long documents and to focus only on matching information, which makes it important.
>
> 2. “It only improves ROUGE-L, BLEU, BLEURT, BertScore, and CHRF, but not ROUGE-1/2 and ANLI.”
>
> As a “sentence matching” function, ROUGE-1/2 and ANLI have concrete flaws. ROUGE-1/2 only matches either unigrams or bigrams, in contrast to BLEU (where all n-grams up to n=4 are considered collectively) and BertScore (where the unigrams are contextualized). ANLI is an NLI classifier that tries to predict binary scores, thus predictions are mostly close to 0 or 1. This is not good for ranking, since two systems with different amounts of errors are given the same score. We further elaborated this in Section A.1 of the current version.
>
> 3. “Ref-only SMART gets extremely bad performance but BLEURT without SMART is not as bad. But with sources added, including the references still largely improves the performance. How would you interpret these results?”
>
> As explained in the Ablation Studies in Section 5.1, the reference summaries used in SummEval are not of good quality – they are rated lower than most of the system summaries in all four dimensions in the original paper [1]. Moreover, SummEval human annotations are done by having only access to source documents. Therefore, it should be expected that correlation scores are bad for reference-only metrics, and those having good correlation scores should be considered erroneous.
>
> Nevertheless, we posit that there still exists a handful of well-written reference summaries, which makes using both the source and the reference in SMART improves the correlation scores. We improved the explanation in the Ablation Studies in the current version.
>
> 4. “How do you justify the idea of using sentences as units in summary evaluation? What about cases where predicted sentences=[“A”, “BC”]; reference sentences=[“AB”, “C”]?”
>
> The original motivation of using sentences as units in summary evaluation is to mitigate problems when the texts are longer, since token-level metrics are not as effective as the text becomes longer (as shown in Figure 1).
>
> To justify the use of sentences better, we added another motivation in the Introduction, where we relate our method to the traditional pyramid-style human annotations in summarization [2]. This annotation technique transforms the reference summary into semantic content units (SCUs), or sentences that represent a single fact in the summary. Humans are then tasked to check whether each SCU is found in the system summary. Transforming a summary into SCUs, however, cannot be automated using current techniques. In our work, we instead use a sentence splitter as a proxy to this transformation.
>
> We also acknowledged the special case mentioned by the reviewer as a limitation in Section 6. We do believe that when we have the technology to transform a text into a list of SCUs, SMART becomes a stronger metric for text generation tasks.
>
> [1] SummEval: Re-evaluating Summarization Evaluation. TACL 2021.
>
> [2] Evaluating Content Selection in Summarization: The Pyramid Method. NAACL 2004.

---

> > ### Comment · Reviewer_7UHv · 2022-11-19
> > **Response**
> >
> > Thank the authors for the detailed explanations and extra results.
> >
> > For 1, my idea was the same as reviewer TC5h: what I wanted to see would be taking the maximum of matching scores for references and source, not source-only, for each sample. In other words, adding the source as one of the reference (that's what I meant by source **inclusion**). From your responses there, it's great to see these results are included, showing that source inclusion is not performing as well as the full SMART method. I think this (source inclusion, instead of source only) is the most important setting to compare with, and should be emphasized more (i.e. try to do this for other methods as well, and put in Table 1; if not having enough space, can probably replace the source only ("src-XXX") rows you added, which are less interesting in my opinion)
> >
> > For other points, I agree with authors' explanations.
> >
> > Overall, I have more confidence in the technical contributions of the paper and will increase the score. However, please continue improving the results presentation in the paper, so that it is easier for readers to get the merits.

---

### Official Review · Reviewer_K7DY · 2022-10-25

**Confidence:** 4
**Correctness:** 4
**Technical Novelty And Significance:** 3
**Empirical Novelty And Significance:** 4
**Recommendation:** 8

**Clarity, Quality, Novelty And Reproducibility:**

The clarity of the work and its contributions can be clearly received from the paper. According to the provided details in the paper most of the results seem to be reproducible. The novelty of the work come from the substitution of metric's base unit with sentences and analyzing different matching metrics' effects on the metric's performance. Also low risk of bias introduced by the metric and its good performance for evaluating long summaries are very beneficial attributes that should not be neglected.

**Strength And Weaknesses:**

The contribution, idea, motivation and results from experiments are written very neatly.

There are quite a comprehensive set of experiments (comparisons to different baselines and functions) to show the effectiveness of the proposed metric.

According to the experiments the metric can be easily adopted to different tasks and domains which can be one of the major positive impact of this metric.

Overall, I do think the paper does not have any major flaws that prevent it from being accepted. Some minor concerns for me is: what was the motivation that authors tried to consider sentences as the units of the evaluation, explaining it can be beneficial. Also, is it possible to show some results of the metric on other reference-based evaluation tasks such as translation?



**Summary Of The Paper:**

This paper is about an automatic reference-based evaluation metric which evaluates the quality of generations with respect to the source text and multiple references by considering sentences as the base units of evaluations and leveraging different matching methods to compute the similarity between sentences. Authors have shown that the proposed metric has better performance even for the evaluation of long summaries and also it is less biased.

**Summary Of The Review:**

This paper is in good shape and the idea behind the metric is interesting and very easily applicable in different domains. The effectiveness of the metric to evaluate long summaries without adding bias is the major benefit of the paper.

---

> ### Author Response · Authors · 2022-11-14
> **Response**
>
> We thank the reviewer for their valuable comments. We updated the paper, addressing the following concerns mentioned in the above review:
>
> 1. “The motivation that the authors tried to consider sentences as the units of the evaluation.”
>
> Our main motivation is based on the idea that tokens as basic units for evaluation do not scale well (in terms of both effectiveness) as the length of the text gets longer. We clearly see the effectiveness of ROUGE and BartScore drop drastically when the summary length is longer in Figure 1.
>
> Our idea is also motivated by traditional pyramid-style human evaluation method [1], where summaries are split into semantic content units (SCUs), each of which is a sentence that represents a single fact mentioned in the summary. We currently do not have an effective way to automatically generate SCUs from a given summary, but splitting it into sentences is a good proxy to this. We added this motivation in the Introduction section, and also mentioned SCU generation in the future work.
>
> 2. “Some results of the metric on other reference-based evaluation tasks such as translation.”
>
> We would like to test SMART on all kinds of text generation tasks including machine translation, however meta-evaluation datasets for machine translation are only with sentence- or segment-level annotations. We believe that SMART can be an effective method to evaluate document-level translations.
>
> [1] Evaluating Content Selection in Summarization: The Pyramid Method. NAACL 2004.

---

### Official Review · Reviewer_TC5h · 2022-10-25

**Confidence:** 4
**Correctness:** 2
**Technical Novelty And Significance:** 3
**Empirical Novelty And Significance:** 3
**Recommendation:** 5

**Clarity, Quality, Novelty And Reproducibility:**

This paper has many points that could be improved in terms of paper quality. Since there are likely many points that can be pointed out and resolved within the authors, it would be desirable to submit a paper in a more complete state. For example:
- If the equations for ROUGE-N and ROUGE-L were shown before SMART-N and SMART-L, the difference (sentence-by-sentence soft match) would become clearer and self-contained would increase.
- In Table 3, rows and columns are inverted. The reader would be easier to read if the methods are always placed on the row-side.
- Without a shorter acronym, the name of the proposed method is much easier to read.

SMART-1 is similar to Sentence Mover's Similarity (Clark et al., ACL 2019); both of which use a soft similarity per sentence. However, there is no experimental comparison or qualitative evaluation, and PROs/CONs are unknown.

**Strength And Weaknesses:**

Strengths:
- Both proposals are intuitive and easy to reproduce.
- The existence of an ablation study (§5.1) makes it easy to understand the nature of the proposed method.

Weaknesses:
- The performance of the proposed method is not robust to the choice of matching function. In fact, on SumEval (§5.1), the proposed method shows high performance with BLEURT, while it shows low performance with ANLI (Table 4). On the other hand, on the TRUE benchmark (§5.2), the proposed method shows high performance with ANLI, while it shows low performance with BLEURT (Table 7).
  - Existing methods should not be described as "very slow" (Introduction); the proposed method also uses LM-based metrics similar to existing methods, e.g., BLEURT. Moreover, the proposed method in this paper has the additional cost that the reader needs to try the number of matching functions for each data set.
- In the paper, the first proposal (sentence-by-sentence computation) is given a large amount of space, but in practice, the second proposal (comparison of generated text and source document) appears to be more effective. In fact, the performance of the first proposal alone is very low (`ref-only` in Table 2). The readers would benefit from an ablation study in which only the second proposal is applied to existing methods such as ROUGE, BLEURT, and ANLI. It would also be very attractive to present concrete examples where the treatment of the sentence-by-sentence works and add qualitative evaluation.
  - Although the authors consider the proposed method as a general-purpose evaluation metric for text generation models, it cannot be applied to machine translation. This is because the proposed method requires calculating the similarity between the source and the candidate. The authors should weaken their argument and argue that they proposed an automatic evaluation metric for automatic summarization. In any case, additional experiments on the TRUE benchmark is very attractive.

**Summary Of The Paper:**

The authors proposed SMART, a new automatic evaluation metric text generation models.
- Proposal 1: To deal with multiple sentences, they proposed SMART-N and SMART-L, in which the calculation unit of ROUGE-N and ROUGE-L is changed from words to sentences. In SMART, several metrics such as BLEURT and ANLI are used to softly compute matching scores between sentences.
- Proposal 2: To exploit the source document, they proposed to use the maximum value of "similarity between candidate text and reference text" and "similarity between candidate text and source text".

Experiments on SummEval (document summarization) and TRUE benchmark (factual consistency) showed that the proposed method performs competitively when using a specific matching function that was selected for each task.

**Summary Of The Review:**

The authors propose a new evaluation metric named SMART for text generation models. Both proposals are intuitive and simple, and cross-sectional experiments suggest empirical effects.

On the other hand, the effectiveness of the main proposal is somewhat questionable, and the paper is not complete; it is considered difficult to be accepted by ICLR.

---

> ### Author Response · Authors · 2022-11-14
> **Response**
>
> We thank the reviewer for their valuable comments. We updated the paper, addressing the following concerns mentioned in the above review:
>
> 1. “SMART is not robust to the choice of matching function per task/dataset. There is an additional cost to try matching functions for each task/dataset.”
>
> Each task has its own set of dimensions to evaluate. In SummEval, the aim is to be able to *rank* systems based on four dimensions (coherence, factuality, grammaticality, and informativeness). BLEURT [1] has been specifically trained to rank texts using objectives that correspond to multiple dimensions of text quality. In TRUE, the goal is to *predict* the amount of factual (in)consistencies in the generated output. ANLI [2] is an NLI classifier that outputs a binary prediction (entailed or not). SMART can be seen as a “smart” way to transform sentence-level metrics into multi-sentence. One would only need to try SMART on the best sentence-level metric for a given task. We clarified this much more clearly in Section 5 of the current version.
>
> 2. “Existing methods should not be described as very slow.”
>
> We have removed this argument from the Introduction section, because although we provided a version that doesn’t use any LM-based model (SMART-CHRF), our best model uses one. We do still believe that SMART-CHRF is a viable metric to use when fast evaluation is necessary (e.g., during the model development phase).
>
> 3. “The first proposal (sentence-by-sentence computation) is given a large amount of space, but in practice, the second proposal (comparison of generated text and source document) appears to be more effective. The readers would benefit from an ablation study in which only the second proposal is applied to existing methods such as ROUGE, BLEURT, and ANLI.”
>
> Thank you for this suggestion. We include source-only versions of ROUGE-1/2/L, CHRF, and BLEURT in Table 1. All of them perform worse than their SMART versions (see general response for a summary of the results).
>
> There are two possible reasons for this. Firstly, the source document is naturally longer than the reference summary, and as also argued in the paper, these metrics are not effective when the documents are longer. Secondly, the source document contains information that does not need to be in the summary. These metrics factor all this excess information, making the scores really low. The sentence-level computation in SMART provides a way to handle both long documents and to focus only on matching information, which makes it important.
>
> 4. “It cannot be applied to machine translation because of source inclusion.”
>
> SMART can be applied to machine translation, even with source inclusion. One could use a multilingual model that allows cross-lingual comparisons. In fact, the BLEURT checkpoint we used is already a multilingual model. When such a model is not available, SMART can be used without including the source as well. We would like to test SMART on all kinds of text generation tasks including machine translation, however meta-evaluation datasets for machine translation are only with sentence- or segment-level annotations. We believe that SMART can be an effective method to evaluate document-level translations.
>
> [1] BLEURT: Learning Robust Metrics for Text Generation. ACL 2020.
>
> [2] TRUE: Re-evaluating Factual Consistency Evaluation. NAACL 2022.

---

> > ### Comment · Reviewer_TC5h · 2022-11-16
> > **Response to authors**
> >
> > Thank you for your thoughtful response! Unfortunately, however, your answer did not resolve my doubts.
> >
> > (1) Presenting a hypothesis (additional description in Sec. 5) after confirming experimental results (Table 1, Table 3) is called HARKing, a research practice that raises doubts about reproducibility. This is because it is doubtful that the same findings would hold when applied to other/future tasks.
> >
> > (3)
> > > Table 1 for baseline metrics (ROUGE, CHRF, and BLEURT) that compare with the source instead of the reference.
> > > (h/t general response)
> >
> > The second proposal by the authors should be [to use the maximum value of "similarity between candidate text and reference text" **and** "similarity between candidate text and source text"], and not [compare with the source **instead of** the reference].
> > This is a very interesting experiment but is not an ablation study in which only the second proposal is applied to existing methods.
> >
> > (4)
> > I can understand that there is no data set to which the first proposal can be easily applied. However, it is non-trivial how to apply the second proposal involving the comparison of the source text (e.g. French text) and the candidate text (e.g. English text) in MT.

---

> > > ### Author Response · Authors · 2022-11-19
> > > **Additional response**
> > >
> > > Thanks again for the follow-up response!
> > >
> > > (1) We disagree that this is harking and that our argument would not hold in other tasks/datasets. For one thing, the added statement in the first paragraph of Section 5 was mentioned after having done the experiments in Sections A.1 and A.5 -- it is an observation of the results from the experiments, not a hypothesis.
> > >
> > > Also, apart from SummEval, we tested different sentence matching functions on arXiv, GovReport, BBC, and CNN, all of which involve ranking systems. On all datasets except CNN, SMART with BLEURT performs the best (we updated Tables 5 and 7 with these results). These are all done after the matching function experiment in Section A.1, which is an evidence that SMART-BLEURT generalizes:
> > > - arXiv: SMART-BLEURT: 0.411, SMART-CHRF: 0.383, SMART-ROUGE (best among 1/2/L): 0.375
> > > - GovReport: SMART-BLEURT: 0.281, SMART-CHRF: 0.271, SMART-ROUGE (best among 1/2/L): 0.210
> > > - BBC: all versions of SMART except BLEURT failed to correctly rank the systems
> > > - CNN: all versions correctly rank the systems
> > >
> > > Finally, the TRUE benchmark consists of multiple datasets from different tasks. We again reiterate that the reason why an NLI classifier is the best sentence matcher for factual consistency is due to the nature of the evaluation task. The task is not to rank systems, but instead to calculate the percentage of factually inconsistent examples, thus the use of ROC AUC in Table 3. The NLI classifier naturally has a good ROC curve as it predicts values close to 0 or 1. To further support this argument, we can see in Table 3 (row 2) that SMART-ANLI performs well in SummEval *when evaluation is using the TRUE setup*, however it underperforms *when evaluation is to rank systems* as in the original SummEval setup (in Table 4).
> > >
> > > (3) We added a row in Table 2 which shows that simply taking the maximum of src-BLEURT and ref-BLEURT (avg 0.528) does not perform as well as SMART-BLEURT (avg 0.628).
> > >
> > > (4) While we think that SMART-BLEURT is suitable for cross-lingual comparisons as BLEURT is a multilingual metric, there is no way to check this as all translation datasets are sentence-level. Therefore, we added in the future work (Section 6.1) that we plan to investigate this.

---

### Official Review · Reviewer_k533 · 2022-10-28

**Confidence:** 4
**Correctness:** 4
**Technical Novelty And Significance:** 3
**Empirical Novelty And Significance:** 3
**Recommendation:** 6

**Clarity, Quality, Novelty And Reproducibility:**

Clarity
The paper is very clearly presented, with adequate details included in the main draft to propose the main ideas. Extra details and results are also neatly presented in the appendix.
Quality
The authors perform adequate experiments to first justify their choices for the soft-matching functions. Thereafter, exhaustive experiments are performed to demonstrate the advantage of the proposed metric in different settings. While I would like the authors to address some questions, overall, I am satisfied with the quality of the presented results.
Novelty
The work is novel. However, the idea of using source-documents for metric calculation has been investigated in the past. Also, given that the model-generated summaries are majorly abstractive, it is not quite clear why using source documents, rather than reference summaries (which are again abstractive in case of CNN/DM), gives better scores especially for quality dimensions other than factuality.
Reproducibility
Reproducing the results could be challenging as codes are not released yet.


**Strength And Weaknesses:**

Strengths
The paper is well-written and well-structured.
Using sentences as basic units of matching, although not new, is well-motivated.
The proposed metric is defined well, although it would be better if the authors could explain SMART-L more clearly, maybe using an example (in the appendix).
Experiments are well-defined and results are clearly presented.
SMART not only outperforms metrics for evaluating content quality of model-generated summaries, but also for evaluating their factual consistency. This gives SMART an added advantage over existing metrics.

Weaknesses
Some results, especially in Table 2, are difficult to follow. Text summarization is more intuitive when it is abstractive as it is closer to how humans summarize a piece of text. Reference summaries on CNN/DM are also abstractive. I believe the majority of the model-generated summaries used for analysis in the SummEval benchmark (Fabbri et al. (2021)) are also abstractive. In such a case, how is source-only SMART performing better than ref-only SMART? It seems counterintuitive to me that abstractive sentences in model-generated summaries would match better with source document sentences. Please explain.
How does using the source documents for SMART calculation improve the correlation scores for non-factuality quality dimensions?
In Table 2, how can the minimum of src-only and ref-only correlation scores be more than the lower of the two? Am I missing something?
Although not a weakness, but can SMART be used in more dynamic settings? For example, consider task-agnostic dialogue generation, where there can be multiple possible responses based on the given context, and also where the response to be generated might depend not only on the immediate context but also on the history of previous utterances.


**Summary Of The Paper:**

The paper proposes a new automatic metric called SMART for evaluating model-generated text. Here, sentences are used as basic units of matching instead of tokens, in order to support long and multi-sentence texts. For this, the authors use a sentence-level soft matching function, and experiment with both string-based and model-based matching functions to select the best variants of SMART. Similar to ROUGE, two types of SMART are proposed, SMART-N which is based on sentence-level n-gram overlap, and SMART-L which is based on longest common soft-subsequence. In order to support grounded evaluation, the authors also include the source documents during metric calculation. Results presented in Table 1 demonstrate that SMART, with BLUERT as the soft-matching function, has better correlation scores, with human judgements, than existing model-based metrics across all four quality dimensions of coherence, fluency, informativeness, and factuality. SMART (SMART-CHRF) also achieves comparable evaluation results with previous LM-based metrics even without using any pretrained language models. Ablation results demonstrate the advantage of using source documents in metric calculation, not only for factuality dimension, but for other dimensions as well. Results presented in the appendix further demonstrate that SMART performs better as summary length increases and is less biased than other competing metrics.


**Summary Of The Review:**

Overall, the paper is a nice read. The proposed metric SMART is well-motivated, well-defined, and shows better correlation scores with human judgements than existing summary-evaluation metrics on four different quality dimensions, namely coherence, fluency, informativeness, and factuality. SMART is flexible, as different soft-matching functions can be selected depending on the task. It also performs well in reference-free settings. The authors demonstrate that SMART performs better when the length of summaries increase. However, in the majority of summarization tasks, summaries are meant to be concise. Hence, this finding may not be practically important. Also, some observations, highlighted under “Weaknesses”, are difficult to follow. I would encourage the authors to address the questions asked in order to make the submission stronger. Overall, the proposed metric is a welcome value addition to the existing literature on model-generated text evaluation metrics.

---

> ### Author Response · Authors · 2022-11-14
> **Response**
>
> We thank the reviewer for their valuable comments. We updated the paper, addressing the following concerns mentioned in the above review:
>
> 1. “How is src-only SMART performing better than ref-only SMART? It seems counterintuitive to me that abstractive sentences in model-generated summaries would match better with source document sentences. Please explain.”
>
> There are multiple reasons for this observation. Firstly, the human annotation of summaries in the SummEval benchmark is done by only looking at the source document. There is no bias towards summaries that look closer to reference summaries. Secondly, the reference summaries in the CNN/DM dataset are article highlights, and are not natural human-written summaries. In fact, one finding in the original SummEval paper [1] shows that CNN/DM summaries are rated lower than all the system summaries by humans. And finally, CNN/DM reference summaries are mostly extractive [2], abstractive models trained on this dataset tend to generate extractive summaries.
>
> Moreover, we further provided evidence that CNN/DM summaries are not good summaries in Table 2, where we compared the correlation score of SMART when using (a) CNN/DM summaries as reference, or (b) the best system summary as reference. The table shows that using the best system summary as reference gives a positive correlation for ref-only SMART.
>
> Due to all these reasons, it is not surprising that src-only SMART performs better than ref-only SMART on the SummEval benchmark. We already had mentioned this in the last paragraph of the Ablation Studies; we made sure to emphasize this in the current version.
>
> 2. “How can the minimum of src- and ref-only SMART have a lower correlation than either src-only and ref-only SMART?”
>
> Intuitively, getting the minimum of src- and ref-only SMART entails that the summary *should* match both the source and the reference. This is a very strict constraint since having only been matched to either source or reference can still be valuable (e.g., less informative but factually consistent sentence). On the other hand, maximum only requires the summary to be matched to either source or reference, which is more robust for multi-dimensional evaluation. We added a sentence that explains this in the current version.
>
> 3. “Consider other task such as task-agnostic dialogue generation”
>
> We would be very happy to try SMART for evaluating different text generation tasks, however most of the available meta-evaluation datasets focus only on short inputs/outputs. Nevertheless, our experiment in Section 5.2 on the TRUE benchmark includes dialogue generation and paraphrase detection, where we also showed improvements when using SMART.
>
> 4. “Reproducing the results could be challenging as codes are not released yet.”
>
> We are planning to share a link to our code after the decisions are announced. For now, we uploaded a zipped file containing our code.
>
> 5. “The authors demonstrate that SMART performs better when the length of summaries increase. However, in the majority of summarization tasks, summaries are meant to be concise. Hence, this finding may not be practically important.”
>
> The usefulness of a summary is in relation to the original document, i.e. the compression ratio. Research has moved toward summarizing longer documents, where longer summaries are produced and deemed as useful. Paper abstracts are longer than news summaries. Synopsis of books and movies are usually multiple paragraphs. Current metrics cannot effectively evaluate these types of summaries.
>
> [1] SummEval: Re-evaluating Summarization Evaluation. TACL 2021.
>
> [2] NEWSROOM: A Dataset of 1.3 Million Summaries with Diverse Extractive Strategies. NAACL 2018.

---

### Author Response · Authors · 2022-11-14
**General Response**

We thank all the reviewers for their time to give valuable feedback and comments to our paper. We have updated the paper (updated parts are highlighted with blue), with the following major revisions:

1. As suggested by reviewer k533 and TC5h, we added correlation results in Table 1 for baseline metrics (ROUGE, CHRF, and BLEURT) that compare with the source instead of the reference. As we expected, these metrics are substantially inferior to their corresponding SMART versions. This shows that the sentence-level computation also helps in improving the performance of SMART. We summarized the result here:

Average correlation

src-ROUGE-1: 0.223
SMART-ROUGE-1: 0.490

src-ROUGE-2: 0.282
SMART-ROUGE-2: 0.432

src-ROUGE-L: 0.323
SMART-ROUGE-L: 0.498

src-CHRF: 0.273
SMART-CHRF: 0.523

src-BLEURT: 0.507
SMART-BLEURT: 0.619

2. We included in the Introduction that this work is also motivated by the traditional pyramid-style summary annotation [1], where the summary is transformed into a list of semantic content units (SCUs), or sentences that contain a single atomic fact. In our work, we use sentence splitting as a proxy to this process. We hope that this, along with the fact the current metrics are not effective for longer summaries, would motivate our work better.

3. To further show that SMART is superior when evaluating long summaries, we compared SMART and several other metrics in a newly curated meta-evaluation suite for long-form document summarization [2], which just got accepted in EMNLP 2022. This experiment shows that in GovReport and arXiv datasets where the summaries are very long, SMART effectively evaluates them in terms of relevance and factuality, while all the other metrics struggle to do so. We added this experiment in Section A.2, and summarized the results here:

Geometric average of correlation with relevance and factuality on GovReport/arXiv:

ROUGE-1/2/L: 0.00 / 0.00

BARTScore: 0.12 / 0.17

SMART: 0.28 / 0.41

[1] Evaluating Content Selection in Summarization: The Pyramid Method. NAACL 2004.

[2] How Far are We from Robust Long Abstractive Summarization? EMNLP 2022.

---

### Decision · Program_Chairs · 2023-01-20

**Decision:**

Accept: poster

**Justification For Why Not Higher Score:**

Intuitive (non-surprising) idea. More comparison is desired.

**Justification For Why Not Lower Score:**

Simple method with good performance.

**Metareview: Summary, Strengths And Weaknesses:**

The paper proposes a new automatic metric, SMART, for evaluating the quality of generated text w.r.t. multiple references and the source text. The metric uses sentences as the base units of evaluations (instead of tokens as commonly used in previous metrics). The metric plugs in existing metrics (e.g., BLEURT) to compute the similarity between sentences. The proposed metric shows better performance for evaluating long-form generations such as summaries. Ablation studies are provided. Overall the idea is simple and intuitive, and the results are positive. Reviewers have suggested more comparisons/analyses, such as the approach of adding source text as one of the references.



**Note From Pc:**

if the above contains the word "oral" or "spotlight" please see: "oral" presentation means -> notable-top-5% and "spotlight" means -> notable-top-25%. As stated in our emails, we are disassociating presentation type from AC recommendations